# Effectiveness and safety of continuous low-molecular-weight heparin versus switching to direct oral anticoagulants in cancer-associated venous thrombosis

Wei Kang [1], Caige Huang[1], Vincent K. C. Yan [1], Yue Wei[1], Jessica J. P. Shami[1], Silvia T. H. Li[1], Yu Yang[1], Xuxiao Ye[1], Junhan Tang[1], Shing Fung Lee [2,3], Victor H. F. Lee [4,5], Stephen L. Chan [6], Aya El Helali[4], Ka On Lam [4,5], Roger K. C. Ngan[4], Ian C. K. Wong [1,7,8,9,10] & Esther W. Chan [1,7,10,11] ✉

Given the existing uncertainty regarding the effectiveness and safety of switching from low-molecular-weight heparin (LMWH) to direct oral anticoagulants (DOACs) in patients with cancer-associated venous thrombosis (CAT), we conducted a comprehensive population-based cohort study utilizing electronic health database in Hong Kong. A total of 4356 patients with CAT between 2010 and 2022 were included, with 1700 (39.0%) patients switching to DOAC treatment. Compared to continuous LMWH treatment, switching to DOACs was associated with a significantly lower risk of hospitalization due to venous thromboembolism (HR: 0.49 [95% CI = 0.35–0.68]) and all-cause mortality (HR: 0.67 [95% CI = 0.61–0.74]), with no significant difference in major bleeding (HR: 1.04 [95% CI = 0.83–1.31]) within six months. These findings provide reassurance regarding the effectiveness and safety of switching from LMWH to DOACs among patients with CAT, including vulnerable patient groups.

Patients with cancer have up to a seven-fold increased risk of venous thromboembolism (VTE) compared to the general population[1,2]. Cancer-associated thrombosis is the second leading cause of death in patients with cancer[3]. Cancer-associated venous thrombosis (CAT) is most commonly associated with an increased risk of recurrent VTE and bleeding[4,5]. Patients with CAT seek medical care more frequently than regular patients with cancer, leading to a two-fold increase in healthcare costs in all settings and a four-fold increase in drug costs, posing a significant financial burden to their families and the healthcare system[6].

Currently, both low-molecular-weight heparin (LMWH) and direct oral anticoagulants (DOACs) are recommended as anticoagulation therapy in patients with CAT[7–9]. Compared to parenteral LMWH, oral

[1]Centre for Safe Medication Practice and Research, Department of Pharmacology and Pharmacy, LKS Faculty of Medicine, The University of Hong Kong, Hong Kong SAR, China. [2]Department of Radiation Oncology, National University Cancer Institute, National University Hospital, Singapore, Singapore. [3]Yong Loo Lin School of Medicine, National University of Singapore, Singapore, Singapore. [4]Department of Clinical Oncology, School of Clinical Medicine, LKS Faculty of Medicine, The University of Hong Kong, Hong Kong SAR, China. [5]Clinical Oncology Center, The University of Hong Kong-Shenzhen Hospital, Shenzhen, China. [6]State Key Laboratory of Translational Oncology, Department of Clinical Oncology, Hong Kong Cancer Institute, The Chinese University of Hong Kong, Hong Kong SAR, China. [7]Laboratory of Data Discovery for Health (D24H), Hong Kong Science and Technology Park, Hong Kong SAR, China. [8]School of Pharmacy, Aston University, Birmingham B4 7ET, England. [9]School of Pharmacy, Medical Sciences Division, Macau University of Science and Technology, Macau SAR, China. [10]The University of Hong Kong Shenzhen Institute of Research and Innovation, Shenzhen, China. [11]Department of Pharmacy, The University of Hong Kong-Shenzhen Hospital, Shenzhen, China. ✉e-mail: ewchan@hku.hk

DOACs were reported to improve compliance with long-term anticoagulation therapy[10]. Previous studies suggested that 51% of LMWH users discontinued treatment within six months after initiation, 21% of whom discontinued due to injection site reactions, compared to 29.5% in the DOAC group[11,12]. Therefore, DOACs may be preferred over LMWH for patients who require a longer duration of anticoagulation therapy.

To date, several clinical trials and real-world cohort studies have shown that DOACs are superior or non-inferior to LMWH in lowering the risk of recurrent VTE. However, whether there is an increased risk of bleeding is debatable[13–21]. The largest Caravaggio trial also reported improved quality of life associated with apixaban compared to dalteparin[16]. Previous studies have directly compared LMWH versus DOACs in patients diagnosed with CAT. In real-world clinical practice, especially in Hong Kong, most patients with newly diagnosed CAT are commonly given LMWH rather than DOACs as initial treatment. However, studies on the effectiveness and safety of switching from LMWH to DOACs are limited. Only one cohort study in Poland reported that extended DOAC treatment was associated with a lower risk of recurrent VTE and a higher risk of major bleeding compared to continuous LMWH treatment, but the findings were not statistically significant due to the small sample size (65 patients on DOACs and 63 patients on LMWH)[21]. Thus, evaluating the safety and effectiveness of switching from LWMH to DOACs versus persistent LMWH use is an important clinical question. The challenge for clinicians is whether to switch patients with CAT from LMWH to DOACs without increasing the risk of recurrent VTE and bleeding.

In this study, we compare the effectiveness and safety associated with continuous use of LMWH versus switching to DOACs in patients newly diagnosed with CAT in Hong Kong between 2010 and 2022. Our results show that switching to DOACs is associated with a reduced risk of hospitalization due to VTE and all-cause mortality in both the short-term (≤ 6 months) and long-term (> 6 months) periods, with no increased risk of major bleeding. Our findings have significant implications for highlighting the importance and guiding clinical decisions of considering this switching strategy in the management of patients with CAT.

## Results

### Patient characteristics
We identified 10,096 patients with newly diagnosed CAT between January 1, 2010, and December 31, 2022 (Fig. 1). After applying the exclusion criteria, 4,356 patients (2381 [54.7%] women; mean [standard deviation, SD] age at CAT diagnosis, 66.3 [13.2] years; mean [SD] follow-up period, 110.6 [69.7] days) were included (Table 1). There were 1700 (39.0%) patients who received LMWH and then switched to DOACs and 2656 (61.0%) who received continuous LMWH therapy. The median (interquartile range [IQR]) duration between initial LMWH treatment and incident CAT were 1 (0–4) days in the switcher group and 2 (0–7) days in the non-switcher group. The median (IQR) duration of anticoagulant use were 108.5 (43-307.2) days (including 6 [3–34] days of initial LMWH treatment) in the switcher group and 30 (11–90) days (including 4 [2–7] days of initial LMWH treatment) in the non-switcher group, where treatment discontinuation was defined as a > 30-day gap between consecutive prescriptions. Compared with the non-switchers, switchers had a higher prevalence of respiratory system cancer, obesity, diabetes, hypertension and hyperlipidemia; were more likely to receive cancer-related drug therapy rather than radiotherapy within three months; had a lower Khorana risk score; and were less likely to have recent surgeries or accident and emergency (A&E) attendances. No significant difference was found for age, sex, cancer metastasis, Charlson Comorbidity Index (CCI), other comorbidities and medications, history of central venous catheter surgery, recent blood transfusion and hospitalization attendance. After inverse probability of treatment weighting (IPTW), baseline characteristics were well-balanced with standardized mean difference (SMD) < 0.1 for all covariates (Table 1).

### Effectiveness outcomes
The incidence of hospitalization due to VTE was 10 and 21 per 100 person-year for patients in the switcher group and the non-switcher group during the six-month follow-up, respectively (Table 2). The switcher group had a significantly lower risk of hospitalization due to VTE than the non-switcher group (hazard ratio [HR]: 0.49 [95% confidence interval, CI = 0.35-0.68]; subdistribution HR [SHR]: 0.58 [95% CI = 0.42–0.80]). The cumulative incidence curve for hospitalization due to VTE also showed a consistent trend (Fig. 2A). The incidence of hospitalization due to deep vein thrombosis (DVT) was 5 and 14 per 100 person-year for patients in the switcher group and the non-switcher group, respectively (Table 2). The incidence of hospitalization due to pulmonary embolism (PE) was 5 and 7 per 100 person-year for patients in the switcher group and the non-switcher group,

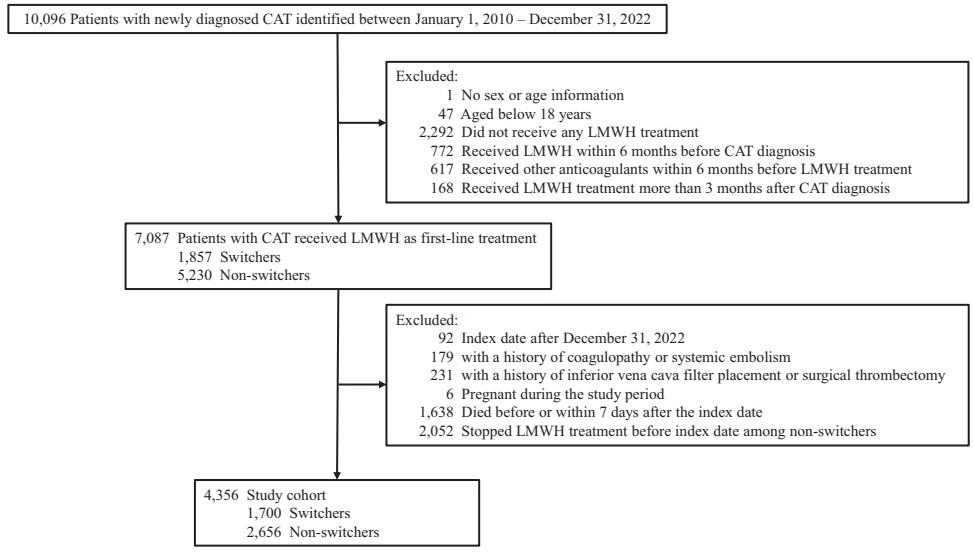

**Fig. 1 | Patient selection flowchart in cohort study.** Cohort selection for analyzing the effectiveness and safety of continuous low-molecular-weight heparin (LMWH) versus switching to direct oral anticoagulants (DOACs) among patients with cancer-associated venous thrombosis (CAT). Switchers were defined as patients who switched to DOACs (apixaban, dabigatran, edoxaban, or rivaroxaban) after receiving LMWH (enoxaparin, tinzaparin, or nadroparin) treatment for any duration. Non-switchers were defined as patients who consistently received LMWH.

**Table 1 | Baseline characteristics of patients with cancer-associated venous thrombosis before and after inverse probability of treatment weighting**

| Characteristics | Unweighted | | | Weighted | | |
|---|---|---|---|---|---|---|
| | Non-Switchers | Switchers | SMD | Non-Switchers | Switchers | SMD |
| N | 2656 | 1700 | | 4347 | 4379 | |
| Age (mean (SD)), year | 65.9 (13.6) | 66.8 (12.6) | 0.069 | 66.3 (13.5) | 66.2 (12.8) | 0.002 |
| Sex, Female | 1420 (53.5) | 961 (56.5) | 0.062 | 2378 (54.7) | 2407 (55.0) | 0.005 |
| Cancer sites | | | | | | |
| Lip oral cavity and pharynx | 50 (1.9) | 28 (1.6) | 0.018 | 77 (1.8) | 75 (1.7) | 0.004 |
| Digestive organs | 773 (29.1) | 447 (26.3) | 0.063 | 1228 (28.3) | 1241 (28.3) | 0.002 |
| Respiratory system | 553 (20.8) | 429 (25.2) | 0.105 | 969 (22.3) | 966 (22.1) | 0.006 |
| Bone skin and soft tissue | 31 (1.2) | 39 (2.3) | 0.086 | 70 (1.6) | 70 (1.6) | < 0.001 |
| Breast and genital organs | 635 (23.9) | 416 (24.5) | 0.013 | 1045 (24.1) | 1049 (23.9) | 0.002 |
| Urinary organs | 125 (4.7) | 60 (3.5) | 0.059 | 184 (4.2) | 179 (4.1) | 0.007 |
| Eye brain and other central nervous system endocrine glands | 35 (1.3) | 21 (1.2) | 0.007 | 55 (1.3) | 57 (1.3) | 0.002 |
| Lymphatic and hematopoietic tissue | 115 (4.3) | 73 (4.3) | 0.002 | 187 (4.3) | 188 (4.3) | < 0.001 |
| Metastasis | 1128 (42.5) | 691 (40.6) | 0.037 | 1813 (41.7) | 1810 (41.3) | 0.008 |
| Comorbidities | | | | | | |
| Obesity | 53 (2.0) | 68 (4.0) | 0.118 | 121 (2.8) | 122 (2.8) | < 0.001 |
| Tobacco use disorder | 16 (0.6) | 21 (1.2) | 0.066 | 32 (0.7) | 34 (0.8) | 0.004 |
| Alcohol use disorder | 30 (1.1) | 20 (1.2) | 0.004 | 50 (1.1) | 52 (1.2) | 0.003 |
| Other drug abuse disorder | 14 (0.5) | 7 (0.4) | 0.017 | 22 (0.5) | 24 (0.5) | 0.006 |
| Diabetes | 391 (14.7) | 315 (18.5) | 0.102 | 699 (16.1) | 701 (16.0) | 0.002 |
| Hypertension | 688 (25.9) | 559 (32.9) | 0.154 | 1241 (28.6) | 1252 (28.6) | 0.001 |
| Hyperlipidemia | 242 (9.1) | 278 (16.4) | 0.219 | 512 (11.8) | 514 (11.7) | 0.001 |
| Atrial fibrillation | 80 (3.0) | 52 (3.1) | 0.003 | 134 (3.1) | 133 (3.0) | 0.002 |
| Congestive heart failure | 109 (4.1) | 61 (3.6) | 0.027 | 168 (3.9) | 172 (3.9) | 0.003 |
| Vascular disease | 172 (6.5) | 110 (6.5) | <0.001 | 283 (6.5) | 288 (6.6) | 0.002 |
| Renal disease | 171 (6.4) | 81 (4.8) | 0.073 | 255 (5.9) | 270 (6.2) | 0.013 |
| Medications in the last three months | | | | | | |
| Cancer-related treatment | | | | | | |
| Drug therapy | 1035 (39.0) | 837 (49.2) | 0.208 | 1867 (42.9) | 1864 (42.6) | 0.007 |
| Radiotherapy | 171 (6.4) | 69 (4.1) | 0.107 | 241 (5.5) | 244 (5.6) | 0.001 |
| Aspirin | 447 (16.8) | 246 (14.5) | 0.065 | 699 (16.1) | 723 (16.5) | 0.012 |
| Antiplatelet (except Aspirin) | 65 (2.4) | 33 (1.9) | 0.035 | 100 (2.3) | 105 (2.4) | 0.007 |
| NSAIDs (except Aspirin) | 522 (19.7) | 283 (16.6) | 0.078 | 805 (18.5) | 822 (18.8) | 0.006 |
| Erythropoietin | 10 (0.4) | 4 (0.2) | 0.026 | 13 (0.3) | 12 (0.3) | 0.006 |
| EGFR inhibitors | 171 (6.4) | 150 (8.8) | 0.090 | 321 (7.4) | 325 (7.4) | 0.002 |
| VEGF/VEGF receptor inhibitors | 67 (2.5) | 45 (2.6) | 0.008 | 113 (2.6) | 111 (2.5) | 0.003 |
| CYP3A4/P-glycoprotein inducers/inhibitors | 299 (11.3) | 169 (9.9) | 0.043 | 469 (10.8) | 479 (10.9) | 0.005 |
| SSRIs/SNRIs | 84 (3.2) | 75 (4.4) | 0.065 | 156 (3.6) | 151 (3.4) | 0.008 |
| Khorana Risk Score | | | 0.164 | | | 0.002 |
| Low (< 2) | 1049 (39.5) | 809 (47.6) | | 1854 (42.7) | 1864 (42.6) | |
| High (>=2) | 1607 (60.5) | 891 (52.4) | | 2493 (57.3) | 2515 (57.4) | |
| CCI | | | 0.039 | | | 0.010 |
| Low (< 6) | 1108 (41.7) | 677 (39.8) | | 1786 (41.1) | 1821 (41.6) | |
| High (> = 6) | 1548 (58.3) | 1023 (60.2) | | 2561 (58.9) | 2558 (58.4) | |
| History of central venous catheter surgery | 74 (2.8) | 73 (4.3) | 0.082 | 148 (3.4) | 148 (3.4) | 0.001 |
| Surgery in the last three months | 1146 (43.1) | 646 (38.0) | 0.105 | 1803 (41.5) | 1827 (41.7) | 0.005 |
| Blood transfusion in the last three months | 164 (6.2) | 75 (4.4) | 0.079 | 240 (5.5) | 252 (5.8) | 0.010 |
| Hospitalization attendance in the last three months (mean (SD)) | 3.1 (2.3) | 3.0 (2.4) | 0.051 | 3.0 (2.3) | 3.1 (2.5) | 0.004 |
| A&E attendance in the last three months (mean (SD)) | 1.5 (1.2) | 1.3 (1.2) | 0.133 | 1.4 (1.2) | 1.4 (1.2) | 0.001 |

*N* Number of patients, *SMD* Standardized mean difference, *SD* Standard deviation, *NSAIDs* Non-steroidal anti-inflammatory drugs, *EGFR* Epidermal growth factor receptor, *VEGF* Vascular endothelial growth factors, *CYP3A4* cytochrome P450 3A4, *SSRIs/SNRIs* Selective serotonin reuptake inhibitors/serotonin and norepinephrine reuptake inhibitors, *CCI* Charlson comorbidity index, *A&E*: Accident and emergency.

**Table 2 | Risk of outcomes associated with switchers vs. non-switchers in patients with cancer-associated venous thrombosis**

| | Events/Incidence (per 100 person-year) | | Unweighted | | Weighted | | Competing Risk | |
|---|---|---|---|---|---|---|---|---|
| | Non-switchers (N = 2656) | Switchers (N = 1700) | Hazard ratio (95% CI) | P value | Hazard ratio (95% CI) | P value | Sub-distribution hazard ratio (95% CI) | P value |
| Hospitalization due to VTE | 148/21 | 53/10 | 0.48 (0.35–0.65) | 3.73e-6 | 0.48 (0.35–0.68) | 1.61e-5 | 0.58 (0.42–0.80) | 7.70e-4 |
| Hospitalization due to DVT | 99/14 | 28/5 | 0.38 (0.25–0.58) | 5.73e-6 | 0.40 (0.26–0.61) | 3.01e-5 | 0.49 (0.32–0.74) | 7.20e-4 |
| Hospitalization due to PE | 52/7 | 26/5 | 0.68 (0.42–1.08) | 0.104 | 0.67 (0.41–1.09) | 0.108 | 0.75 (0.46-1.25) | 0.280 |
| Major bleeding | 192/27 | 139/26 | 1.01 (0.81-1.26) | 0.931 | 1.04 (0.83-1.31) | 0.724 | 1.18 (0.94-1.48) | 0.150 |
| ICH | 24/3 | 11/2 | 0.63 (0.31-1.29) | 0.205 | 0.63 (0.30-1.32) | 0.223 | 0.72 (0.35-1.51) | 0.390 |
| GI bleeding | 48/6 | 41/7 | 1.17 (0.77-1.78) | 0.458 | 1.23 (0.80-1.90) | 0.338 | 1.43 (0.93-2.19) | 0.100 |
| Bleeding of other critical sites | 122/17 | 89/16 | 1.02 (0.78-1.35) | 0.865 | 1.05 (0.79-1.40) | 0.728 | 1.18 (0.89-1.56) | 0.260 |
| All-cause mortality | 1629/216 | 729/129 | 0.61 (0.56-0.67) | 6.53e-28 | 0.67 (0.61-0.74) | 1.63e-17 | - | - |

N Number of patients, CI Confidence interval, VTE Venous thromboembolism, DVT deep vein thrombosis, PE pulmonary embolism, ICH intracranial hemorrhage, GI bleeding gastrointestinal bleeding. All P values were from 2-sided tests and results were deemed statistically significant at P < 0.05.

respectively. Switching to DOACs was associated with a significant reduction in hospitalization due to DVT (HR: 0.40 [95% CI = 0.26–0.61]; SHR: 0.49 [95% CI = 0.32–0.74]) and no increase in hospitalization due to PE (HR: 0.67 [95% CI = 0.41–1.09]; SHR: 0.75 [95% CI = 0.46–1.25]).

### Safety outcomes

The incidence of major bleeding was 26 and 27 per 100 person-year for the switcher group and the non-switcher group during the six-month follow-up, respectively (Table 2). There was no significant difference in the risk of major bleeding between the two groups (HR: 1.04 [95% CI = 0.83–1.31]; SHR: 1.18 [95% CI = 0.94–1.48]). The cumulative incidence curve for hospitalization due to major bleeding also showed a consistent trend (Fig. 2B). No significant difference was observed for the risk of intracranial hemorrhage (ICH) (HR: 0.63 [95% CI = 0.30–1.32]; SHR: 0.72 [95% CI = 0.35–1.51]), gastrointestinal bleeding (GI bleeding) (HR: 1.23 [95% CI = 0.80–1.90]; SHR: 1.43 [95% CI = 0.93–2.19]), and bleeding of other critical sites (HR: 1.05 [95% CI = 0.79–1.40]; SHR: 1.18 [95% CI = 0.89–1.56]) (Table 2).

Within six months of follow-up, there were 729 deaths (42.9%) in the switcher group compared to 1629 (61.3%) among non-switchers (Table 2). The incidence of all-cause mortality was 129 and 216 per 100 person-year for patients in the switcher group and the non-switcher group, respectively. There was a significantly lower risk of all-cause mortality in the switcher group (HR: 0.67 [95% CI = 0.61–0.74]). The cumulative incidence curve for all-cause mortality also showed a consistent trend (Fig. 2C).

### Case validation

We performed case validation for a total of 435 (10%) patients (Supplementary Fig. 1). The positive predictive values (PPVs) of active cancer, hospitalization due to VTE, and major bleeding were 99.8%, 90.3%, and 95.2%, respectively. The negative predictive values (NPVs) for hospitalization due to VTE and major bleeding were 98.8% and 99.5%, respectively.

Among 28 true positive cases of hospitalization due to VTE, 21 (75.0%) received a computerized tomography scan or Doppler ultrasound confirming hospitalization for recurrent VTE cases, 5 (17.9%) were hospitalized because of worsening VTE-related symptoms (such as localized pain, swelling, and mobility difficulties), and 2 (7.1%) were hospitalized for other VTE-related reasons. Among these true positive cases, 4 cases were from the switcher group (3 recurrent VTE cases and 1 VTE-related symptom case), and 24 cases were from the non-switcher group (18 recurrent VTE cases, 4 VTE-related symptom cases, and 2 other cases).

### Subgroup analyses

A significant reduction in hospitalization due to VTE was observed in most switcher subgroups compared to non-switchers, except for patients with gastrointestinal malignancy or platinum chemotherapy within three months before the index date, among whom no significant difference was detected (Table 3). No differences were found in major bleeding except switchers with lower bleeding risk, among whom significant increase was detected (Table 3). Consistent results were found in all subgroup analyses for all-cause mortality except no significant difference in platinum chemotherapy within three months before the index date.

### Sensitivity analyses

Results were shown in Supplementary Table 1–9. Generally, the results of the sensitivity analyses were consistent with those of the main analyses (Supplementary Table 1–4, 6–9), except when the index year was considered a confounder, there was no significant difference in the risk of hospitalization due to VTE (HR: 0.50 [95% CI = 0.31–0.80]; SHR: 0.78 [95% CI = 0.53–1.14]) and DVT (HR: 0.36 [95% CI = 0.21–0.61]; SHR:

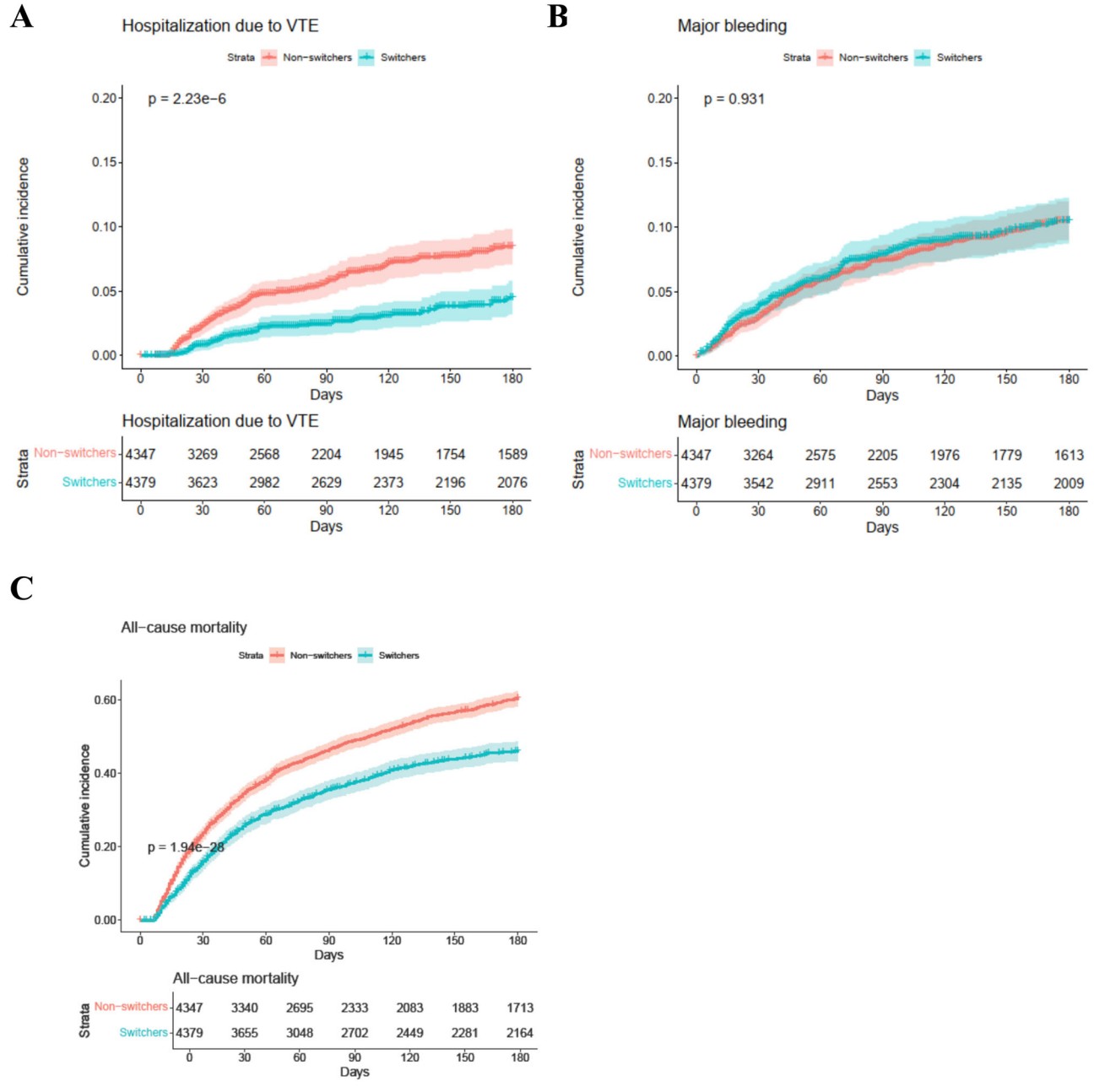

**Fig. 2 | Cumulative incidence of each outcome in patients with cancer-associated venous thrombosis.** The cumulative incidence for each outcome was compared between the two groups using IPTW-weighted Kaplan-Meier survival analysis and log-rank test. **A** Cumulative incidence of hospitalization due to venous thromboembolism (VTE) in patients with cancer-associated venous thrombosis (CAT) within a 6-month follow-up period. **B** Cumulative incidence of major bleeding in patients with CAT within a 6-month follow-up period. **C** Cumulative incidence of all-cause mortality in patients with CAT within a 6-month follow-up period. Blue line represents switchers ($N = 4379$), patients who switched to DOACs after receiving LMWH treatment for any duration. Pink line represents non-switchers ($N = 4347$), patients who consistently received LMWH. Censored data points were represented as "+". Error bands represent the 95% confidence intervals for each effect estimate. *P*-values were shown for two-sided log-rank test comparisons and results were deemed statistically significant at $P < 0.05$.

0.75 [95% CI = 0.46–1.22]) among switchers and non-switchers (Supplementary Table 5).

As shown in Supplementary Fig. 2, after applying the exclusion criteria, a total of 4877 patients with CAT between 2010 and 2022 were included. Among them, 1051 patients switched to DOAC treatment, and 3826 received continuous LMWH therapy within the initial 30 days after CAT diagnosis. The results in Supplementary Table 7 are consistent with the main analysis. Compared with persistent use of LMWH, switching to DOACs was associated with a significantly lower risk of hospitalization due to VTE (HR: 0.54 [95% CI = 0.33–0.89]; SHR: 0.60 [95% CI = 0.39–0.94]) and all-cause mortality (HR: 0.61 [95%

CI = 0.51–0.73]) with no difference in major bleeding (HR: 0.98 [95% CI = 0.68–1.41]; SHR: 1.08 [95% CI = 0.75–1.56]) within six months. In sensitivity analysis 9 (Supplementary Table 9), the median (IQR) durations of anticoagulant treatment were 78 (48.5–110.5) days in the switcher group and 64.5 (42–104) days in the non-switcher group.

## Discussion

In this large population-based cohort study of patients with CAT, compared to patients on continuous LMWH, those who were switched to DOACs were associated with a significantly lower risk of hospitalization due to VTE and all-cause mortality, with non-inferior major

**Table 3 | Subgroup analysis for risk of outcomes associated with switchers vs. non-switchers in patients with cancer-associated venous thrombosis in six months**

| Subgroups | | Non-Switchers (N) | Switchers (N) | Hospitalization due to VTE | | Major bleeding | | All-cause mortality | |
|---|---|---|---|---|---|---|---|---|---|
| | | | | Adjusted Hazard Ratio (95% CI) | P value | Adjusted Hazard Ratio (95% CI) | P value | Adjusted Hazard Ratio (95% CI) | P value |
| Sex | Male | 1236 | 739 | 0.51 (0.32–0.82) | 0.005 | 0.91 (0.65–1.27) | 0.563 | 0.67 (0.58–0.76) | 5.08e-9 |
| | Female | 1420 | 961 | 0.49 (0.31–0.77) | 0.002 | 1.20 (0.88–1.64) | 0.256 | 0.69 (0.61–0.78) | 2.21e-9 |
| Age | <=65 years | 1274 | 764 | 0.51 (0.32–0.82) | 0.006 | 0.83 (0.58–1.18) | 0.294 | 0.72 (0.62–0.83) | 3.99e-6 |
| | >65 years | 1382 | 936 | 0.46 (0.30–0.72) | 7.11e-4 | 1.22 (0.90–1.65) | 0.198 | 0.64 (0.57–0.73) | 1.28e-12 |
| Metastasis | No | 1528 | 1009 | 0.58 (0.40–0.86) | 0.006 | 1.23 (0.91–1.67) | 0.178 | 0.60 (0.53–0.69) | 2.11e-14 |
| | Yes | 1128 | 691 | 0.33 (0.17–0.63) | 7.42e-4 | 0.87 (0.62–1.23) | 0.438 | 0.77 (0.67–0.87) | 5.95e-5 |
| Gastrointestinal malignancy | No | 1883 | 1253 | 0.49 (0.33–0.70) | 1.43e-4 | 1.08 (0.84–1.40) | 0.552 | 0.66 (0.59–0.74) | 1.40e-13 |
| | Yes | 773 | 447 | 0.51 (0.26–1.00) | 0.052 | 0.90 (0.55–1.49) | 0.692 | 0.72 (0.61–0.85) | 1.12e-4 |
| Platinum treatment within 3 months before index date | No | 2296 | 1453 | 0.46 (0.32–0.65) | 1.06e-5 | 1.09 (0.85–1.39) | 0.507 | 0.65 (0.59–0.7) | 4.40e-18 |
| | Yes | 360 | 247 | 0.79 (0.33–1.89) | 0.600 | 0.85 (0.46–1.55) | 0.589 | 0.89 (0.70–1.13) | 0.351 |
| CCI | Low (<6) | 1108 | 677 | 0.59 (0.35–0.98) | 0.042 | 1.25 (0.87–1.81) | 0.231 | 0.62 (0.53–0.72) | 2.25e-9 |
| | High (>=6) | 1548 | 1023 | 0.42 (0.27–0.64) | 5.06e-5 | 0.93 (0.69–1.25) | 0.628 | 0.71 (0.64–0.80) | 3.80e-9 |
| Khorana risk score | Low (<2) | 1049 | 809 | 0.49 (0.31–0.79) | 0.003 | 1.11 (0.78–1.60) | 0.557 | 0.67 (0.57–0.78) | 1.22e-7 |
| | High (>=2) | 1607 | 891 | 0.45 (0.29–0.71) | 6.46e-4 | 1.01 (0.75–1.37) | 0.955 | 0.68 (0.61–0.77) | 1.95e-10 |
| Bleeding risk | Low | 718 | 517 | 0.44 (0.23–0.84) | 0.013 | 1.68 (1.06–2.65) | 0.026 | 0.57 (0.48–0.69) | 5.97e-9 |
| | High | 1938 | 1183 | 0.50 (0.34–0.72) | 2.44e-4 | 0.91 (0.70–1.19) | 0.486 | 0.71 (0.64–0.79) | 1.74e-10 |
| Treatment duration before switching | Shorter (<=10 days) | 2656 | 935 | 0.57 (0.38–0.85) | 0.007 | 1.06 (0.80–1.41) | 0.673 | 0.72 (0.64–0.81) | 3.04e-8 |
| | Longer (>10 days) | 2656 | 765 | 0.37 (0.22–0.60) | 6.39e-5 | 1.06 (0.77–1.47) | 0.704 | 0.64 (0.56–0.74) | 3.70e-10 |

N Number of patients, CI Confidence interval, VTE Venous thromboembolism, CCI Charlson comorbidity index. All P values were from 2-sided tests and results were deemed statistically significant at $P < 0.05$.

bleeding during both the six-month follow-up (short-term) and one-year follow-up periods (long-term). The significant reduction in hospitalization due to VTE was consistent with previous randomized controlled trials (RCTs) and cohort studies comparing LMWH versus DOACs[14,15,18,20]. In a single-center prospective cohort study in Poland, no statistically significant lower risk of recurrent VTE (HR 0.44 [95% CI = 0.16–1.16]) and higher risk of major bleeding (HR 2.00 [95% CI = 0.50–8.00]) was found for switchers compared to non-switchers[21].

Increased bleeding risk is the main concern when deciding whether to initiate/extend anticoagulation therapy for patients with CAT, especially in Asian populations that are at higher risk of bleeding with DOACs compared to other populations[22,23]. Data addressing the bleeding risk associated with LMWH and DOACs lacks consensus and remains controversial[14,17,19,21]. To date, there are no cohort studies comparing the risk of ICH between DOACs and LMWH, although ICH is the most serious major bleeding type side-effect that threatens life span and is associated with a heavier health management burden than non-ICH major bleeding[24,25]. Importantly, given the large sample size, our study has sufficient power to detect and address minor differences in bleeding risk. The results indicate that switching to DOACs was associated with non-inferior risk of ICH, GI bleeding and bleeding of other critical sites compared to continuous use of LMWH.

The risk of mortality in real-world clinical practice between the two groups differs significantly, and although our findings are not consistent with published RCTs, they are consistent with previous real-world cohort studies[13,15–17,20]. Patients in the RCTs are usually under intensive monitoring in hospital settings during the whole study period. In contrast, patients in the real world may have been discharged from hospital and do not receive the same level of intensive care. Further, RCTs generally exclude vulnerable populations, patients with poor health status or prognosis after VTE, and those with contraindications to the study medications.

Considering the potential impact of COVID-19 on the healthcare system, we conducted a sensitivity analysis to address any influence on the frequency of hospital visits, treatment preference, and disease progress during this period. Our study conclusions are consistent and are not influenced by the COVID-19 pandemic. It is important to note that our patient cohort was recruited between 2010 and 2022, during which DOACs became available in Hong Kong, and clinical practice for cancer management developed over this period. Patients recruited at the earlier time points were much less likely to receive DOACs due to inaccessibility and lack of evidence on the safety and clinical benefits of DOACs in CAT. Thus, we considered the index year as a potential confounder. The sensitivity analysis results remained consistent with the main analysis with no statistically significant reduction for hospitalization due to VTE and DVT among switchers when considering death as a competing event. Besides, given that some non-switchers may have deceased before reaching their pseudo index date and that potential immortal time bias could be introduced, we conducted a landmark analysis to ensure that patients in both groups have an equal opportunity to be included in the study. The sensitivity analysis results remained consistent with the main analysis, indicating that the conclusion was robust and not confounded by the immortal time bias.

In addition, edoxaban and dabigatran users might have received a five- to ten-day LWMH treatment prior to the use of DOACs. Therefore, we further separated the switcher groups into shorter switchers (treatment duration before switching ≤10 days) and longer switchers (treatment duration before switching >10 days) and compared them to non-switchers respectively. Both shorter switchers and longer switchers were associated with a significantly lower risk of hospitalizations due to VTE and all-cause mortality, with non-inferior major bleeding. Notably, we observed that in real-world clinical practice, the median total duration of anticoagulant treatment in the non-switcher group was much shorter than that in the switcher group. The shorter LMWH treatment duration could be due to treatment discontinuation

due to side effects, or transfer to palliative care. Thus, our findings likely reflect the intention-to-treat effect of switching to DOACs versus continuing LMWH for the prevention of recurrent VTE in CAT. We further conducted a sensitivity analysis to censor patients at treatment discontinuation to evaluate the per-protocol effect of switching to DOACs versus continuing LMWH, and the results were robust.

Apart from the beneficial outcomes of switching to DOACs, patient preference is also an important consideration. The COSIMO study reported that patients strongly preferred oral administration compared to the parenteral route of administration, which influences long-term adherence to therapy and clinical outcomes[26,27].

### Strengths and limitations

To our knowledge, this is the first population-based study to evaluate the association between switching to DOACs or continuous use of LMWH and the risk of hospitalization due to VTE, major bleeding, and all-cause mortality among patients with CAT. Second, this is the first cohort study to evaluate ICH, which has not been comparatively investigated between DOAC and LMWH users. Third, this study provided essential insight into the treatment outcomes of some vulnerable populations, including patients who underwent recent platinum chemotherapy, poorer performance status, higher VTE and/or bleeding risk. Last but not least, we assessed the accuracy of healthcare database records, including coding of active cancer as well as diagnosis records for hospitalization due to VTE and major bleeding, and explored the reasons for hospitalization due to VTE. Most population-based studies on CAT used the concept of active cancer, which has no corresponding the International Classification of Diseases, Ninth Revision, Clinical Modification (ICD-9-CM) code and did not perform case validation[28–30]. Our validation results suggest that our strategy for identifying patients with active cancer was precise, thus ensuring accuracy in our outcome findings.

This study has several limitations. (1) Some clinical confounders such as the severity of the diseases and cancer staging are not directly available in the Clinical Data Analysis and Reporting System (CDARS) database. However, we calculated the CCI and Khorana risk score to predict 10-year survival and risk of VTE and used IPTW to minimize baseline confounding. (2) Due to the lack of adherence information, discrepancies might exist between the prescription records obtained from CDARS and the actual medication use. This is an inherent problem in most real-world studies. However, during case validation, we found that patients with CAT attended clinician appointments frequently and actively discussed the treatment strategy with clinicians. As a result, adherence issues should be minimal. (3) Since 54.1% of patients had died within six months after diagnosis, the cumulative incidence of hospitalization due to VTE and major bleeding might have been overestimated without counting death as a competitive event. However, the consistency between results generated using Cox regression and the Fine-Gray sub-distribution hazard model showed that the competing risk did not bias our results. (4) Due to the limited sample size, we could not evaluate the outcomes for each specific medication in either anticoagulant class. Future studies are needed to generalize our findings for each DOAC. (5) Anticoagulant dosage was not considered in this study. A robust method to analyze the effectiveness and safety of different dosages of LMWH and DOACs should be developed for future studies. (6) Although our findings suggest that switching to DOACs is beneficial for patients with CAT on a population level, the decision to switch to DOACs after initial LMWH therapy still requires evaluation of the risks and benefits on an individualized basis.

In conclusion, compared with the continuous use of LMWH, switching to DOACs was associated with a lower risk of hospitalization due to VTE and all-cause mortality both short-term (≤ 6 months) and long-term (>6 months), with no increase in major bleeding among patients with CAT. Our study provides reassurance that switching to DOACs may be considered in patients with CAT after initial LMWH

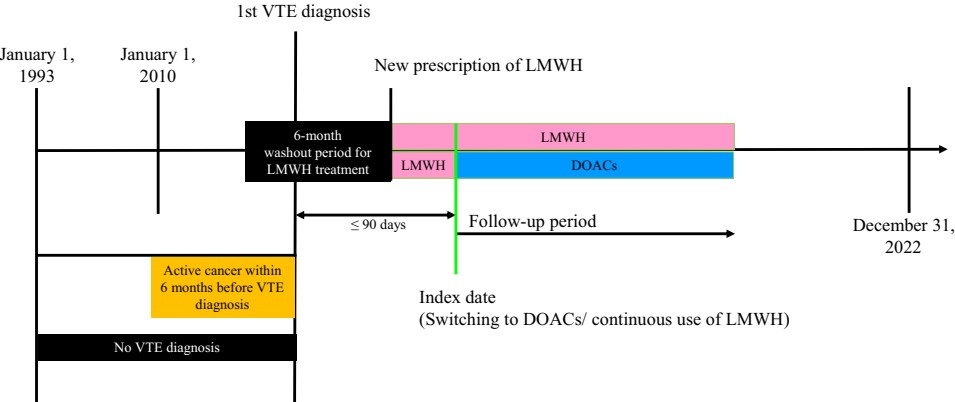

**Fig. 3 | Design of cohort study.** Patients with cancer-associated venous thrombosis (CAT) were identified as those with active cancer followed by an incident diagnosis of venous thromboembolism (VTE) between January 1, 2010, and December 31, 2022. Active cancer (yellow band) was defined as a new cancer diagnosis, recurrent cancer diagnosis, presence of metastasis, recent cancer-related treatment, or palliative care within the six months prior to the first VTE diagnosis. Switchers were defined as patients who switched from receiving low-molecular-weight heparin (LMWH, pink band) treatment to direct oral anticoagulants (DOACs, blue band). Non-switchers were patients who consistently received LMWH treatment (pink band). The start of initial LMWH treatment was determined as the date of the first recorded LMWH prescription following the 1st VTE diagnosis. Only patients who received LMWH treatment within 90 days after the CAT diagnosis were included in this study. Patients with a history of LMWH treatment within six months prior to the initial LMWH treatment were excluded as LMWH-experienced. For the switcher group, the index date was defined as the first date of DOAC treatment. A pseudo-index date was randomly assigned based on the distribution (mean ± standard deviation [SD]) of the time difference between the commencement of LMWH treatment and the switch to DOACs in the switcher group. Patients were followed up from the index date (green line) for six months or until the earliest occurrence of the outcome, death, or the end of the study period (December 31, 2022).

therapy, specifically in the elderly, metastasis cancer, gastrointestinal malignancy, platinum-therapy regimen, poorer performance status, and those with higher VTE and/or bleeding risk. Our findings provide real-world evidence that switching from LMWH to DOACs is a safe and effective strategy in the management of CAT.

## Methods

This study was approved by the Institutional Review Board of The University of Hong Kong/Hospital Authority Hong Kong West Cluster (Reference Number: UW 20-697). This is an anonymized pharmacoepidemiology study without patient contact or any personal identification information. Therefore, informed consent was infeasible to obtain and thus exempted. This cohort study follows the Strengthening the Reporting of Observational Studies in Epidemiology reporting guidelines[31].

### Data sources

This study used electronic health records from the CDARS database provided by the Hong Kong Hospital Authority[32]. In 2022, 43 public hospitals and 122 outpatient clinics were under the Hospital Authority management, serving over 7.9 million people[33]. Anonymized patient data from the CDARS database include demographic information, diagnoses, prescription records, hospitalization records, outpatient clinic attendance, A&E attendance, procedures, and laboratory tests, which have been widely used in previous studies[34–37].

### Study population

Patients with CAT, defined as patients with active cancer followed by an incident diagnosis of VTE between January 1, 2010, and December 31, 2022, were identified. A new cancer diagnosis, recurrent diagnosis of cancer, metastasis, any cancer-related treatment, or palliative care in the last six months before the first VTE diagnosis were considered as active cancer[38]. As shown in Fig. 1, the exclusion criteria were as follows: patients without sex or age information, under 18 years, did not receive any LMWH (enoxaparin, tinzaparin, or nadroparin) treatment, received LMWH within six months before CAT diagnosis, received other anticoagulants within six months before LMWH treatment, or started LMWH treatment later than three months after diagnosis of CAT.

### Study design

Figure 3 shows the study design concept. Patients who were switched to DOACs (apixaban, dabigatran, edoxaban, or rivaroxaban) after LMWH treatment of any duration were considered as the switcher group. Patients who continuously received LMWH were considered as the non-switcher group. For the switcher group, the index date was defined as the first date of DOAC treatment. Based on the distribution (mean ± SD) of the time difference between commencement of LMWH treatment and switching to DOACs in the switcher group, a pseudo-index date was randomly assigned to the non-switcher group[39]. As shown in Fig. 1, patients who were assigned an index date after December 31, 2022, had a history of coagulopathy or systemic embolism, had a history of inferior vena cava filter placement or surgical thrombectomy, were pregnant during the study period, died before or within seven days after the index date, or discontinued LMWH treatment before the index date were excluded. All diagnoses and procedures were identified using ICD-9-CM codes (Supplementary Table 10). All the medications used were identified using the British National Formulary codes (Supplementary Table 11).

### Outcomes

The primary outcome of interest was hospitalization due to VTE, including DVT and PE. Hospitalization due to VTE within 14 days after the diagnosis of CAT was not considered as an outcome since they were likely part of the same CAT episode. Secondary outcomes were major bleeding (including ICH, GI bleeding, and bleeding of other critical sites) and all-cause mortality[17]. Major bleeding was defined based on a list of ICD-9 codes (Supplementary Table 10), considering both primary and secondary diagnoses, and encompassing events from A&E, inpatient, and outpatient settings. Patients were followed up from the index date for six months or until the earliest occurrence of the outcome, death, or the end of the study period (December 31, 2022).

### Case validation

An independent clinical oncologist manually validated and confirmed identification of active cancer as well as the outcomes of hospitalization due to VTE and major bleeding. This was achieved by comparing CDARS records to medical notes from the Clinical Management

System for a randomly selected patient population, comprising 10% of the entire cohort[40]. PPV and NPV were calculated for data validation.

## Statistical analysis

Baseline patient characteristics were presented as mean (SD) for continuous variables and frequencies (percentages) for categorical variables. IPTW using propensity score was performed to minimize potential confounding and baseline differences between the two groups. Covariates included in calculating the propensity score were age, sex, cancer sites, metastasis, comorbidities (including obesity, tobacco use disorder, alcohol use disorder, other drug use disorder, diabetes, hypertension, hyperlipidemia, atrial fibrillation, congestive heart failure, vascular disease, and renal disease), cancer-related treatment (including drug therapy and radiotherapy) within three months before the index date, concomitant medication use within three months before the index date (including aspirin, antiplatelets, non-steroidal anti-inflammatory drugs [NSAIDs], erythropoietin, epidermal growth factor receptor [EGFR] inhibitors, vascular endothelial growth factors [VEGF]/VEGF receptor inhibitors, cytochrome P450 3A4 inducers/inhibitors, P-glycoprotein inducers/inhibitors, and selective serotonin reuptake inhibitors [SSRIs]/serotonin and norepinephrine reuptake inhibitors [SNRIs]), CCI (Supplementary Table 12)[37,41], Khorana risk score[42], history of central venous catheter surgeries, any surgeries within three months before the index date, blood transfusion within three months before the index date, hospitalization, A&E attendances within three months before the index date (Supplementary Table 10 and 11). A SMD of < 0.1 after weighting was considered acceptable.

The cumulative incidence for each outcome was compared between the two groups using IPTW-weighted Kaplan-Meier survival analysis and log-rank test. The association of hospitalization due to VTE, major bleeding, and all-cause mortality between switchers and non-switchers was estimated using IPTW-weighted Cox proportional hazards regression with a robust variance estimator. Results were reported as HR with 95% CIs. All P values were from 2-sided tests and results were deemed statistically significant at $P < 0.05$. Considering death as a competing risk, the Fine-Gray sub-distribution hazard model was also employed to estimate the outcome risks.

Subgroup analyses were performed by sex, age (≤ 65 or > 65 years), metastasis (yes or no), cancer site (gastrointestinal malignancy or not), cancer-related treatment (underwent platinum chemotherapy within three months before the index date, which may cause a higher risk of VTE), CCI (< 6 or ≥ 6), Khorana risk score (< 2 or ≥ 2), bleeding risk (high or low), and treatment duration before switching (≤ 10 or > 10 days). The high bleeding risk group was composed of patients with history of renal disease, liver disease, regional/advanced metastatic cancer, history of major bleeding within three months before the index date, history of surgery within two weeks before the index date, or receiving medications that increase the risk of bleeding (including aspirin, antiplatelets, NSAIDs, thrombolytics, EGFR inhibitors, VEGF/VEGF receptor inhibitors) within six weeks before the index date.

Nine sensitivity analyses were performed in this study: (1) the follow-up period was extended to one year to explore the long-term effectiveness and safety; (2) patients with basal cell carcinoma, squamous-cell skin carcinoma, lymphoma, acute leukemia, or myeloproliferative neoplasm were excluded because they may have a different risk profile in terms of CAT[43,44]; (3) patients with a short follow-up period (index date later than July 4, 2022) were excluded to ensure that each patient had the same observational period; and (4) patients diagnosed with CAT on or after January 1, 2020 were excluded because the COVID-19 pandemic may have influenced treatment decisions regarding anticoagulant therapy and hospitalization; (5) the index year was included as a confounder because the DOAC availability and cancer management might improve over time; (6) the end date of anticoagulation treatment was added as one of the follow-up

endpoints; (7) A landmark analysis was conducted to address the potential immortal time bias. Supplementary Fig. 3 showed the concept of this study design. The 30th day following the incident CAT was set as the landmark. After applying the same exclusion criteria for patients with newly diagnosed CAT between January 1, 2010, and December 31, 2022 (Supplementary Fig. 2), the cohort was divided into two groups during the initial 30-day treatment following the incident CAT: patients who had switched to DOACs after LMWH treatment were considered as the switcher group; patients who continuously received LMWH were considered as the non-switcher group. Both groups were followed up from the index date (day 30 after the incident CAT) for a period of six months or until the occurrence of the outcome, death, the end of anticoagulation treatment, or the end of the study period (December 31, 2022), whichever occurred first; (8) a multiple imputation method was used to investigate the implication of missing outcome data for the effect estimate[45]. Twenty imputed datasets were generated with a predictive distribution based on the covariates included in calculating the propensity score as above. A model of interest was fitted in each dataset and combined to give an overall estimate. Results in all twenty imputed datasets were pooled using Rubin's rules[46]; (9) excluding patients whose anticoagulation treatment duration was < 30 days or ≥ 180 days to evaluate the effectiveness and safety between two groups with the same treatment duration.

## Reporting summary

Further information on research design is available in the Nature Portfolio Reporting Summary linked to this article.

## Data availability

The source data supporting the findings described in this manuscript and supplementary information are provided by the data custodians under approval due to concerns over patient privacy protection. Requests for data access can be submitted to the Central Panel on Administrative Assessment of External Data Requests of the Hospital Authority (hacpaaedr@ha.org.hk). The time required to process such requests may vary according to the specific purpose of each project. Factors including personal data privacy issues, the relevance of the requested data to the study purpose, data availability, data limitations, and the resources involved in extracting the data will be considered when assessing each data request. For further information, please refer to the official website of the Hospital Authority's data-sharing portal (https://www3.ha.org.hk/data).

## Code availability

R 4.0.3 (R Foundation for Statistical Computing, Vienna, Austria) was used for all statistical analyses. The analyses were conducted by W.K. and cross-checked independently by C.H. for quality assurance. The code used to generate the results reported in this study is available on GitHub (https://github.com/Faery0105/cohort-study/blob/main/cohort%20study_HK_CAT_20240618.R).

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

## Acknowledgements

This study was supported by the Hong Kong Research Grants Council, General Research Fund (GRF), No. 17108621, which was received by E.W.C. The funding source had no role in the design and conduct of the study, collection, management, analysis, and interpretation of the data. We thank the Hong Kong Hospital Authority for access to the data from the Clinical Database Analysis and Reporting System for research purposes and the Information Technology Services, The University of Hong Kong, for offering research computing facilities for the computations in this study. We thank Lisa Y Lam, MJ, Department of Pharmacology and Pharmacy, The University of Hong Kong, for proofreading the manuscript. Ms. Lam was compensated for her contribution.

## Author contributions

Conception and design: W.K., V.K.C.Y., and E.W.C. Acquisition, analysis, or interpretation of data: W.K., C.H., V.K.C.Y., and E.W.C. Drafting of the manuscript: W.K. Critical revision of the manuscript for important intellectual content: W.K., C.H., V.K.C.Y., Y.W., J.J.P.S., S.T.H.L., Y.Y., X.Y., J.T., S.F.L., V.H.F.L., S.L.C., A.E.H., K.O.L., R.K.C.N., I.C.K.W., and E.W.C. Statistical analysis: W.K. and C.H.; Administrative, technical, or material support: E.W.C. Supervision: E.W.C. and I.C.K.W. All authors reviewed and approved the final manuscript.

## Competing interests

Both I.C.K.W. and E.W.C. received research funding outside the submitted work from Pfizer and Bayer. The other authors declare no competing interests.
