## [Peer Review File · Nature Communications]

Effectiveness and safety of continuous low-molecular-weight heparin versus switching to direct oral anticoagulants in cancer-associated venous thrombosisREVIEWER COMMENTS

Reviewer #1 (Remarks to the Author):

I appreciate the opportunity to review Kang et al.'s manuscript on switching to DOAC therapy for patients with CAT initially treated with LMWH following incident VTE. The manuscript addressed an important topic and is well-written. However, I have some concerns regarding the comparison of switchers vs. non-switchers and whether this is valid and unbiased.

My comments and questions are provided below:

My main concern pertains to the potential presence of immortal time bias. While I acknowledge the authors' attempt to mitigate this by introducing a pseudo-index date for non-switchers, I remain unconvinced that this fully addresses the issue, as patients still need to survive a certain period to receive a pseudo index date. To best circumvent this bias, I would recommend considering participant cloning as proposed by Hernán (How to estimate the effect of treatment duration on survival outcomes using observational data. *BMJ* 2018; 360: k182. 10.1136/bmj.k182). Another approach could involve conducting a landmark analysis, for instance, initiating follow-up at day 120 after the incident VTE and classifying patients based on whether they had switched treatment prior to day 120. I think this would provide a more robust assessment of treatment effects.

Another major concern relates to the substantial disparity in treatment duration between switchers and non-switchers. This discrepancy in adherence, particularly with lower adherence observed among LMWH-continuers, could potentially account for the significant lower rate of recurrent VTE seen in switchers. The authors contend that these duration differences may have minimal impact, as the sensitivity analysis censoring patients at treatment discontinuation yielded findings consistent with the main analysis. However, I have reservations about the validity of this conclusion, as the censoring process is likely not independent (e.g., potentially informative censoring). To mitigate this, the authors might consider employing inverse probability of censoring weights to address potential bias arising from patient discontinuation of anticoagulation.

Even with the application of IPTW, I am still concerned about the potential influence of confounding by indication. The decision to switch to a DOAC is fundamentally driven by clinical judgment, a factor that IPTW cannot entirely account for. I am keen to hear the authors' thoughts on this.

Methods: For the safety outcomes of major bleeding, please include more information on who this was defined, e.g. was this based on both primary and secondary diagnosis, and was both inpatient and outpatient visits considered? This may have implications for the validity.

Line 144 Case Validation: The authors to be commended their rigorous approach in conducting case validation. Could you please specify whether this validation was carried out for all patients or only a random sample? This detail should be clarified in the methods section. Regarding the description of the findings in the results, I find it somewhat difficult to fully comprehend the results, particularly in the description of the PPV for recurrent VTE diagnoses. In Figure S2, the PPV for recurrent VTE is reported as 28 out of 31, which equals 90.3%. However, in the text (lines 250-253), it mentions 21+5+3, totaling 29 patients. It's noted that 5 had worsening of VTE-related symptoms; was this defined as recurrent VTE?

Table 1: I am surprised about the large proportion of patients with a CCI score >6. Could you please provide details on how the CCI score was calculated, and confirm if cancer was excluded from this calculation? Additionally, I note the absence of information regarding the duration of initial LMWH use among switchers, as well as the time elapsed since the index VTE event for both groups. I believe this information is crucial for assessing the comparability of the two exposure groups.

Table 2: please clarify whether the rates presented in the table is pr 100 person-years.

The discussion section appears lengthy and would benefit from some shortening to enhance coherence and avoid redundancy. For instance, I recommend that the authors provide a concise

summary of the study's main findings in the initial paragraph, while avoiding repetition of the introductory material where the rationale is already well-established. For example, lines 272-277 and 335-337 could be omitted. This approach would help maintain focused and nuanced discussion.

Minor comments:

Line 103: ... under HA management.. Please define this abbreviation at first mention.

Line 113: The authors could consider describing the study population and exposure of patients with CAT under a separate heading. Thus, letting the first paragraph pertain only to data sources and approvals. I think this could improve readability. Figure S1 = helpful to understand how the study population is derived.

Reviewer #2 (Remarks to the Author):

This is a well-done analysis of a cohort of patients in Hong Kong treated with LMWH (continuously) vs switching over to DOACs. The authors find that rates of hospitalization were lower in those who switched vs those who did not. These findings support current clinical practice of greater use of DOACs in the cancer population and the findings of several large recent RCTs.

A major issue with this paper (as acknowledged by authors) is that these are findings of correlation. Given that switch to DOACs is amore recent years practice, this finding could simply be that hospitalization rates have fallen over time as cancer treatments have improved. Alternatively, pts being switched to DOACs are healthier than those remaining on LMWHs.

Point-by-point response to reviewers' comments

Reviewer #1 (Remarks to the Author):

I appreciate the opportunity to review Kang et al.'s manuscript on switching to DOAC therapy for patients with CAT initially treated with LMWH following incident VTE. The manuscript addressed an important topic and is well-written. However, I have some concerns regarding the comparison of switchers vs. non-switchers and whether this is valid and unbiased.

My comments and questions are provided below:

Q1. My main concern pertains to the potential presence of immortal time bias. While I acknowledge the authors' attempt to mitigate this by introducing a pseudo-index date for non-switchers, I remain unconvinced that this fully addresses the issue, as patients still need to survive a certain period to receive a pseudo index date. To best circumvent this bias, I would recommend considering participant cloning as proposed by Hernán (How to estimate the effect of treatment duration on survival outcomes using observational data. BMJ 2018; 360: k182. 10.1136/bmj.k182). Another approach could involve conducting a landmark analysis, for instance, initiating follow-up at day 120 after the incident VTE and classifying patients based on whether they had switched treatment prior to day 120. I think this would provide a more robust assessment of treatment effects.

Reply: Thank you for your valuable feedback. We agree that addressing the issue of immortal time bias is crucial to our study. Thus, we conducted a landmark analysis to mitigate the immortal time bias concern. According to the median (IQR) duration between the incident of CAT and switching from LMWH to DOACs in switchers (11 [5-43] days), we designated the 30th day following the incident of CAT as a landmark. As shown in Figure S1 (Supplementary material: page 1), our cohort was then divided into two groups based on whether they switched to DOAC treatment from LMWH during the initial 30 days following the incident CAT or not. Patients were then followed up from the index date (day 30 after the incident CAT) for six months or until the occurrence of the outcome, death, the end of anticoagulation treatment, or the end of the study period (December 31, 2022), whichever occurred first.

We included this analysis as sensitivity analysis 7 in our study and the results are listed in Table S10 (Supplementary material: page 14). In general, the findings were consistent with our main analysis, which

supports the robustness of our results. We discussed the results in the discussion section to demonstrate that our conclusions were not impacted by the potential immortal time bias.

Thank you once again for your input, and we are grateful for your thoughtful review of our manuscript.

We have revised our manuscript as follows:

- Methods (Line 10-20, Page 10): “(7) A landmark analysis was conducted to address the potential immortal time bias. Figure S1 showed the concept of this study design. The 30th day following the incident CAT was set as the landmark. After applying the same exclusion criteria for patients with newly diagnosed CAT between January 1, 2010, and December 31, 2022 (Figure S2), the cohort was divided into two groups during the initial 30-day treatment following the incident CAT: patients who had switched to DOACs after LMWH treatment were considered as the switcher group; patients who continuously received LMWH were considered as the non-switcher group. Both groups were followed up from the index date (day 30 after the incident CAT) for a period of six months or until the occurrence of the outcome, death, the end of anticoagulation treatment, or the end of the study period (December 31, 2022), whichever occurred first;”
- Results (Line 10-11, Page 14): “Generally, the results of the sensitivity analyses were consistent with those of the main analyses (Table S4-S7, S9-S12),”
- Results (Line 15-22, Page 14): “As shown in Figure S2, after applying the exclusion criteria, a total of 4,877 patients with CAT between 2010 and 2022 were included. Among them, 1,051 patients switched to DOAC treatment, and 3,826 received continuous LMWH therapy within the initial 30 days after CAT diagnosis. The results in Table S10 are consistent with the main analysis. Compared with persistent use of LMWH, switching to DOACs was associated with a significantly lower risk of hospitalization due to VTE (HR: 0.54 [95% CI 0.33-0.89]; SHR: 0.60 [95% CI 0.39-0.94]) and all-cause mortality (HR: 0.61 [95% CI 0.51-0.73]) with no difference in major bleeding (HR: 0.98 [95% CI 0.68-1.41]; SHR: 1.08 [95% CI 0.75-1.56]) within six months.”
- Discussion (Line 16-21, Page 17): “Besides, given that some non-switchers may have deceased before reaching their pseudo index date and that potential immortal time bias could be introduced, we conducted a landmark analysis to ensure that patients in both groups have an equal opportunity to be included in the study. The sensitivity analysis results remained consistent with the main analysis, indicating that the conclusion was robust and not confounded by the immortal time bias.”
- Supplementary material (Page 1): “Figure S1. Landmark analysis design”

- Supplementary material (Page 2): “Figure S2. Patient selection flowchart in landmark analysis”

- Supplementary material (Page 14): “Table S10. Sensitivity analysis: The landmark analysis”

Outcomes	Events/Incidence (per 100 person-year)		Unweighted		Weighted		Competing Risk	
	Non-switchers (N=3,826)	Switchers (N=1,051)	Hazard Ratio (95% CI)	P value	Hazard Ratio (95% CI)	P value	Subdistribution Hazard Ratio (95% CI)	P value
Hospitalization due to VTE	92/30	25/13	0.57 (0.37-0.89)	<0.05	0.54 (0.33-0.89)	<0.05	0.60 (0.39-0.94)	<0.05
Hospitalization due to DVT	59/19	12/6	0.43 (0.23-0.79)	<0.05	0.35 (0.18-0.68)	<0.05	0.43 (0.23-0.80)	<0.05
Hospitalization due to PE	35/11	14/7	0.85 (0.46-1.58)	0.60	0.90 (0.45-1.79)	0.77	0.93 (0.48-1.81)	0.83
Major bleeding	93/30	48/26	1.07 (0.76-1.52)	0.70	0.98 (0.68-1.41)	0.92	1.08 (0.75-1.56)	0.66

ICH	6/2	1/1	0.35 (0.04-2.87)	0.32	0.31 (0.04-2.62)	0.28	0.33 (0.02-4.34)	0.40
GI bleeding	22/7	14/7	1.31 (0.67-2.56)	0.43	1.27 (0.64-2.52)	0.49	1.53 (0.74-3.14)	0.25
Bleeding of other critical sites	65/21	34/18	1.09 (0.72-1.65)	0.69	0.96 (0.62-1.49)	0.87	1.05 (0.68-1.61)	0.84
All-cause mortality	579/181	170/88	0.61 (0.51-0.72)	<0.05	0.61 (0.51-0.73)	<0.05	-	-

VTE: venous thromboembolism; DVT: deep vein thrombosis; PE: pulmonary embolism; ICH: intracranial hemorrhage; GI bleeding: gastrointestinal bleeding

Q2. Another major concern relates to the substantial disparity in treatment duration between switchers and non-switchers. This discrepancy in adherence, particularly with lower adherence observed among LMWH-continuers, could potentially account for the significantly lower rate of recurrent VTE seen in switchers. The authors contend that these duration differences may have minimal impact, as the sensitivity analysis censoring patients at treatment discontinuation yielded findings consistent with the main analysis. However, I have reservations about the validity of this conclusion, as the censoring process is likely not independent (e.g., potentially informative censoring). To mitigate this, the authors might consider employing inverse probability of censoring weights to address potential bias arising from patient discontinuation of anticoagulation.

Reply: Thank you for your comment. We acknowledge there is considerable treatment duration difference between these two groups. Therefore, we conducted both intention-to-treat and per-protocol analysis in our study. To address the issue of potentially informative censoring, we conducted an additional analysis using the multiple imputation method proposed by Jackson et al.¹ The multiple imputation method was used to assess the sensitivity of the inferences made from Cox proportional hazards models to relax the independent censoring assumption.^{1,2} The multiple imputation method has been widely used by previous high-quality publications to impute plausible outcomes with their follow-up time distributions for censored data based on available information.³⁻⁵ This process considers the relationship between the censoring mechanism and the outcome variable, helping to minimize potential bias arising from patient discontinuation of anticoagulation. We included this analysis as sensitivity analysis 8 in our study and the results are shown in Table S11 (Supplementary material Page 15). The findings were consistent with our main analysis, which confirmed the assumption of noninformative censoring and support the robustness of our results.

We have revised our manuscript as follows:

- Methods (Line 20, Page 10 to Line 2, Page 11): “(8) a multiple imputation method was used to investigate the implication of missing outcome data for the effect estimate.³⁶ Twenty imputed datasets were generated with a predictive distribution based on the covariates included in calculating the propensity score as above. A model of interest was fitted in each dataset and

combined to give an overall estimate. Results in all twenty imputed datasets were pooled using Rubin’s rules;⁶”

- Results (Line 10-11, Page 14): “Generally, the results of the sensitivity analyses were consistent with those of the main analyses (Table S4-S7, S9-S12),”
- Supplementary material (Page 15): “Table S11. Sensitivity analysis: Results of multiple imputation analysis”

Outcomes	Hazard Ratio (95% CI)	P value
Hospitalization due to VTE	0.45 (0.29-0.70)	<0.05
Hospitalization due to DVT	0.43 (0.24-0.77)	<0.05
Hospitalization due to PE	0.50 (0.24-1.01)	0.05
Major bleeding	1.02 (0.74-1.42)	0.89
ICH	1.59 (0.35-7.15)	0.54
GI bleeding	1.07 (0.58-1.95)	0.83
Bleeding of other critical sites	0.99 (0.66-1.48)	0.95
All-cause mortality	0.52 (0.46-0.58)	<0.05

VTE: venous thromboembolism; DVT: deep vein thrombosis; PE: pulmonary embolism; ICH: intracranial hemorrhage; GI bleeding: gastrointestinal bleeding

Q3. Even with the application of IPTW, I am still concerned about the potential influence of confounding by indication. The decision to switch to a DOAC is fundamentally driven by clinical judgment, a factor that IPTW cannot entirely account for. I am keen to hear the authors' thoughts on this.

Reply: Thank you for your insightful comments. Based on our review of clinical notes and discussions with healthcare professionals, we found that patients' treatment regimens were indeed influenced by clinical judgment. Additionally, discussions with patients regarding the switch to DOACs for convenience or to address bleeding or other adverse events played a role in the decision-making process. These factors, which are inherently based on individual patient factors and clinical considerations, could not be fully balanced in our study. We acknowledge that indication bias cannot be totally eliminated in observational studies, but we used IPTW to reduce this bias. In the previous version of the manuscript, we had already discussed this as a limitation (1) “Some clinical confounders such as the severity of the diseases and cancer staging are not directly available in the CDARS database. However, we calculated the CCI and Khorana risk score to predict 10-year survival and risk of VTE and used IPTW to minimize baseline confounding.” (Line 10-13, Page 19) in our manuscript. Specifically, we have also stated that

“Although our findings suggest that switching to DOACs is beneficial for patients with CAT on a population level, the decision to switch to DOACs after initial LMWH therapy still requires evaluation of the risks and benefits on an individualized basis.” in limitation (6) (Line 3-6, Page 20).

Q4. Methods: For the safety outcomes of major bleeding, please include more information on who this was defined, e.g. was this based on both primary and secondary diagnosis, and was both inpatient and outpatient visits considered? This may have implications for the validity.

Reply: Thank you for your suggestion. We have revised our manuscript accordingly: “Major bleeding was defined based on a list of ICD-9 codes (Table S1), considering both primary and secondary diagnoses, and encompassing events from A&E, inpatient, and outpatient settings.” (Line 22, Page 7 to Line 1, Page 8)

Q5. Line 144 Case Validation: The authors to be commended their rigorous approach in conducting case validation. Could you please specify whether this validation was carried out for all patients or only a random sample? This detail should be clarified in the methods section. Regarding the description of the findings in the results, I find it somewhat difficult to fully comprehend the results, particularly in the description of the PPV for recurrent VTE diagnoses. In Figure S2, the PPV for recurrent VTE is reported as 28 out of 31, which equals 90.3%. However, in the text (lines 250-253), it mentions 21+5+3, totaling 29 patients. It's noted that 5 had worsening of VTE-related symptoms; was this defined as recurrent VTE?

Reply: Thank you for your valuable feedback. We apologize for any confusion caused. To clarify, the case validation was conducted on a random sample comprising 10% of our entire cohort. We have now included this information in the Methods section to provide transparency in our study design: “An independent clinical oncologist manually validated and confirmed identification of active cancer as well as the outcomes of hospitalization due to VTE and major bleeding. This was achieved by comparing CDARS records to medical notes from the Clinical Management System for a randomly selected patient population, comprising 10% of the entire cohort.” (Line 4-7, Page 8).

Regarding the discrepancy in the reported results, we acknowledge the typo in the manuscript. The correct statement should have been: " Among 28 true positive cases of hospitalization due to VTE, 21 (75.0%) received a computerized tomography scan or Doppler ultrasound confirming hospitalization for recurrent VTE cases, 5 (17.9%) were hospitalized because of worsening VTE-related symptoms (such as

localized pain, swelling, and mobility difficulties), and 2 (7.1%) were hospitalized for other VTE-related reasons."(Line 16-20, Page 13) instead of "3 (7.1%) were hospitalized for other VTE-related reasons". The cases of hospitalizations due to VTE included cases of hospitalizations due to recurrent VTE, hospitalizations due to worsening VTE-related symptoms, and hospitalizations for other VTE-related reasons. Therefore, the cases with worsening VTE-related symptoms and cases with recurrent VTE were mutually exclusive in our study. For case validation, we randomly selected 435 cases from our electronic medical records and validated their actual diagnoses recorded in medical notes. Among them, 31 were recorded with hospitalizations due to VTE in electronic medical records, and 28 were recorded with hospitalizations due to VTE in the medical notes. Therefore, the positive predictive value (PPV) is $28/31 = 90.3\%$. Among the 28 true positive cases of hospitalization due to VTE, 21 were hospitalized due to recurrent VTE cases, 5 were hospitalized due to worsening VTE-related symptoms, and 2 were hospitalized for other VTE-related reasons.

In our population-based study, 17.9% of cases with VTE-related worsening symptoms were considered as hospitalization due to VTE. Therefore, we named the outcome as “hospitalization due to VTE” rather than “recurrent VTE”, which is commonly used by other related real-world studies, however the electronic medical records from their databases were not validated. Although the actual recurrent VTE rate was 75%, lower than expected, we endeavored to represent progress in bridging the gap between database studies and clinical practice (including but not limited to removing patients deceased within seven days after switching in both groups to reduce other impacts, rather than solely anticoagulants, ignoring hospitalization due to VTE cases within 14 days after the diagnosis of CAT to avoid duplicated VTE episode, and case validation). We believe our study contributes to a more comprehensive understanding of hospitalization due to VTE and recurrent VTE compared to previous related real-world studies.

Q6. Table 1: I am surprised about the large proportion of patients with a CCI score >6. Could you please provide details on how the CCI score was calculated, and confirm if cancer was excluded from this calculation? Additionally, I note the absence of information regarding the duration of initial LMWH use among switchers, as well as the time elapsed since the index VTE event for both groups. I believe this information is crucial for assessing the comparability of the two exposure groups.

Reply: Thank you for your interest in our study. Regarding the CCI calculation, we have included a list of the specific conditions and corresponding scores in Table S3 (Supplementary material, Page 7) for

clarification. In our study, we included cancer in the CCI calculation because metastatic solid cancer is assigned a score of 6, while other cancer types are assigned a score of 2. Therefore, it is important to consider cancer and the presence of metastasis as risk factors in our analysis.

In response to your concern about the duration of time elapsed in the results, we have included this in the results: “The median (interquartile range [IQR]) duration between initial LMWH treatment and incident CAT were 1 (0-4) days in the switcher group and 2 (0-7) days in the non-switcher group. The median (IQR) duration of anticoagulant use were 108.5 (43-307.2) days (including 6 [3-34] days of initial LMWH treatment) in the switcher group and 30 (11-90) days (including 4 [2-7] days of initial LMWH treatment) in the non-switcher group, where treatment discontinuation was defined as a >30-day gap between consecutive prescriptions.” (Line 16-21, Page 11). We added sensitivity analysis 9, excluding patients whose anticoagulation treatment duration was <30 days or ≥ 180 days to evaluate the safety and effectiveness between the two groups under the same treatment duration. The median (IQR) durations of medication anticoagulant use were 78 (48.5-110.5) days in the switcher group and 64.5 (42-104) days in the non-switcher group. As shown in Table S12 (Supplementary material, Page 16), the findings were consistent with our main analysis, suggesting our results were robust when comparing patients with similar treatment duration.

We have revised our manuscript as follows:

- Methods (Line 1, Page 9): “Charlson Comorbidity Index (CCI, Table S3),”.
- Methods (Line 2-4, Page 11): “(9) excluding patients whose anticoagulation treatment duration was <30 days or ≥ 180 days to evaluate the effectiveness and safety between two groups with the same treatment duration.”
- Results (Line 16-21, Page 11): “The median (interquartile range [IQR]) duration between initial LMWH treatment and incident CAT were 1 (0-4) days in the switcher group and 2 (0-7) days in the non-switcher group. The median (IQR) duration of anticoagulant use were 108.5 (43-307.2) days (including 6 [3-34] days of initial LMWH treatment) in the switcher group and 30 (11-90) days (including 4 [2-7] days of initial LMWH treatment) in the non-switcher group, where treatment discontinuation was defined as a >30-day gap between consecutive prescriptions.”
- Results (Line 10-11, Page 14): “Generally, the results of the sensitivity analyses were consistent with those of the main analyses (Table S4-S7, S9-S12),”
- Results (Line 22, Page 14 to Line 2, Page 15): “In sensitivity analysis 9 (Table S12), the median (IQR) durations of anticoagulant treatment were 78 (48.5-110.5) days in the switcher group and 64.5 (42-104) days in the non-switcher group.”

- Supplementary material (Page 7): “Table S3. Charlson Comorbidity Index calculation list”

Factors	Specify	Score
Age	50-59	+1
	60-69	+2
	70-79	+3
	>=80	+4
Myocardial infarction	Yes	+1
Congestive heart failure	Yes	+1
Peripheral vascular disease	Yes	+1
Cerebrovascular accident or transient ischemic attack	Yes	+1
Dementia	Yes	+1
Chronic obstructive pulmonary disease	Yes	+1
Connective tissue disease	Yes	+1
Peptic ulcer disease	Yes	+1
Liver disease	Mild	+1
	Moderate to severe	+3
Diabetes mellitus	Uncomplicated	+1
	End-organ damage	+2
Hemiplegia	Yes	+2
Moderate to severe chronic renal disease	Yes	+2
Solid tumor	Localized	+2
	Metastatic	+6
Leukemia	Yes	+2
Lymphoma	Yes	+2
Acquired immune deficiency syndrome	Yes	+6

- Supplementary material (Page 16): “Table S12. Sensitivity analysis: Excluding patients whose anticoagulation treatment duration <30 days or ≥180 days”

Outcomes	Events/Incidence (per 100 person-year)		Unweighted		Weighted		Competing Risk	
	Non-switchers (N=1,010)	Switchers (N=763)	Hazard Ratio (95% CI)	P value	Hazard Ratio (95% CI)	P value	Subdistribution Hazard Ratio (95% CI)	P value
Hospitalization due to VTE	74/27	29/14	0.52 (0.34-0.79)	<0.05	0.53 (0.34-0.82)	<0.05	0.56 (0.36-0.86)	<0.05
Hospitalization due to DVT	45/16	13/6	0.38 (0.21-0.71)	<0.05	0.39 (0.21-0.73)	<0.05	0.41 (0.22-0.78)	<0.05
Hospitalization due to PE	30/11	17/8	0.77 (0.42-1.39)	0.38	0.76 (0.41-1.40)	0.38	0.78 (0.41-1.47)	0.44
Major bleeding	87/32	72/35	1.12 (0.82-1.54)	0.46	1.05 (0.76-1.45)	0.75	1.07 (0.78-1.48)	0.67
ICH	11/4	3/1	0.37 (0.10-1.31)	0.12	0.35 (0.10-1.30)	0.12	0.41 (0.10-1.74)	0.23
GI bleeding	20/7	23/11	1.56 (0.86-2.85)	0.14	1.55 (0.84-2.86)	0.16	1.51 (0.78-2.94)	0.22
Bleeding of other critical sites	56/20	47/23	1.14 (0.77-1.67)	0.52	1.02 (0.68-1.52)	0.93	1.00 (0.66-1.51)	0.99
All-cause mortality	710/248	445/208	0.84 (0.75-0.95)	<0.05	0.84 (0.74-0.95)	<0.05	-	-

VTE: venous thromboembolism; DVT: deep vein thrombosis; PE: pulmonary embolism; ICH: intracranial hemorrhage; GI bleeding: gastrointestinal bleeding

Q7. Table 2: please clarify whether the rates presented in the table is pr 100 person-years.

Reply: Thank you for your suggestion. We made changes for all results from person-years to 100 person-years.

We have revised our manuscript as follows:

- Results (Line 7-9, Page 12): “The incidence of hospitalization due to VTE was 10 and 21 per 100 person-year for patients in the switcher group and the non-switcher group during the six-month follow-up, respectively (Table 2).”
- Results (Line 11-15, Page 12): “The cumulative incidence curve for hospitalization due to VTE also showed a consistent trend (Figure 3). The incidence of hospitalization due to DVT was 5 and 14 per 100 person-year for patients in the switcher group and the non-switcher group, respectively (Table 2). The incidence of hospitalization due to PE was 5 and 7 per 100 person-year for patients in the switcher group and the non-switcher group, respectively.”
- Results (Line 20-21, Page 12): “The incidence of major bleeding was 26 and 27 per 100 person-year for the switcher group and the non-switcher group during the six-month follow-up, respectively (Table 2).”
- Results (Line 6-8, Page 13): “The incidence of all-cause mortality was 129 and 226 per 100 person-year for patients in the switcher group and the non-switcher group, respectively.”
- Table 2 (Page 35)
- Table S4-S9 (Supplementary material Page 8-13)

Q8. The discussion section appears lengthy and would benefit from some shortening to enhance coherence and avoid redundancy. For instance, I recommend that the authors provide a concise summary of the study's main findings in the initial paragraph, while avoiding repetition of the introductory material where the rationale is already well-established. For example, lines 272-277 and 335-337 could be omitted. This approach would help maintain focused and nuanced discussion.

Reply: Thank you for your valuable suggestion. We have revised the discussion section according to your recommendation.

We have revised our manuscript as follows:

- Discussion (Line 2-10, Page 16): “In this large territory-based cohort study of patients with CAT, compared to patients on continuous LMWH, those who were switched to DOACs were associated with a significantly lower risk of hospitalization due to VTE and all-cause mortality, with non-inferior major bleeding during both the six-month follow-up (short-term) and one-year follow-up periods (long-term). The significant reduction in hospitalization due to VTE was consistent with previous randomized controlled trials (RCTs) and cohort studies comparing LMWH versus DOACs.^{14,15,18,20} In a single-center prospective cohort study in Poland, no statistically significant lower risk of recurrent VTE (HR 0.44 [95% CI 0.16-1.16]) and higher risk of major bleeding (HR 2.00 [95% CI 0.50-8.00]) was found for switchers compared to non-switchers.⁸”
- Discussion (Line 13-17, Page 16): “Data addressing the bleeding risk associated with LMWH and DOACs lacks consensus and remains controversial.⁸⁻¹¹ To date, there are no cohort studies comparing the risk of ICH between DOACs and LMWH, although ICH is the most serious major bleeding type side-effect that threatens life span and is associated with a heavier health management burden than non-ICH major bleeding.^{12,13}”
- Discussion (Line 13-16, Page 18): “Apart from the beneficial outcomes of switching to DOACs, patient preference is also an important consideration. The COSIMO study reported that patients strongly preferred oral administration compared to the parenteral route of administration, which influences long-term adherence to therapy and clinical outcomes.^{14,15}”

Minor comments:

Q9. Line 103: ... under HA management.. Please define this abbreviation at first mention.

Reply: Thank you for your comments. We have made changes in the manuscript: “In 2022, 43 public hospitals and 122 outpatient clinics were under the Hospital Authority management, serving over 7.9 million people.” (Line 4-6, Page 6)

Q10. Line 113: The authors could consider describing the study population and exposure of patients with CAT under a separate heading. Thus, letting the first paragraph pertain only to data sources and approvals. I think this could improve readability. Figure S1 = helpful to understand how the study population is derived.

Reply: Thank you for your suggestion. Based on your recommendation, we have restructured the method section. Additionally, we have moved the content from the original Figure S1 into the main manuscript as new Figure 1 to provide a clearer understanding of how the study population is designed.

We have revised our manuscript as follows:

- Methods (Line 2, Page 6): “Data sources”
- Methods (Line 15, Page 6): “Study population”
- Methods (Line 1, Page 7): “Study design”
- Methods (Line 2, Page 7): “Figure 1 shows the study design concept.”
- Figure 1 (Page 30): “Figure 1. Design of cohort study”

Reviewer #2 (Remarks to the Author):

This is a well-done analysis of a cohort of patients in Hong Kong treated with LMWH (continuously) vs switching over to DOACs. The authors find that rates of hospitalization were lower in those who switched vs those who did not. These findings support current clinical practice of greater use of DOACs in the cancer population and the findings of several large recent RCTs.

A major issue with this paper (as acknowledged by authors) is that these are findings of correlation. Given that switch to DOACs is a more recent years practice, this finding could simply be that hospitalization rates have fallen over time as cancer treatments have improved. Alternatively, pts being switched to DOACs are healthier than those remaining on LMWHs.

Reply: Thank you for your comments and we understand your concerns. Undeniably, with advances in cancer management, hospitalization rates among patients with cancer have declined over time. However, it is important to note that the hospitalization events studied in this paper specifically refer to hospitalizations with VTE as the primary reason, rather than encompassing all hospitalizations of patients with cancer. Furthermore, we took the index year into consideration as a variable in sensitivity analysis 5, and the results were consistent with the main analysis. Additionally, according to the comparison of baseline characteristics between the two groups, it was found that switchers were not simply healthier than non-switchers. In fact, switchers had a higher prevalence of comorbidities, including respiratory system cancer, obesity, diabetes, hypertension, and hyperlipidemia, when compared to non-switchers.

However, as previously mentioned in the discussion section, the lack of clinical confounders such as disease severity and cancer staging in the CDARS database means that it is still possible for patients who switched to DOACs to have better overall health conditions compared to those who continued with LMWH. To address this concern, we took the following steps: 1) We calculated the Charlson Comorbidity Index (CCI) and Khorana risk score to predict 10-year survival and the risk of VTE, respectively. 2) We implemented measures in our experimental design to balance the baseline characteristics of patients and minimize differences between the two groups. 3) Additionally, we employed the Fine-Gray sub-distribution hazard model to estimate the risk of outcomes while considering death as a competing risk.

These results were consistent with our main analysis. Thus, we believe that within the existing experimental design, we have made every effort to account for confounding factors, and our results are robust.

- 1 Jackson, D. *et al.* Relaxing the independent censoring assumption in the Cox proportional hazards model using multiple imputation. *Stat Med* **33**, 4681-4694 (2014). <https://doi.org/10.1002/sim.6274>
- 2 Zhao, Y., Herring, A. H., Zhou, H., Ali, M. W. & Koch, G. G. A multiple imputation method for sensitivity analyses of time-to-event data with possibly informative censoring. *J. Biopharm. Stat.* **24**, 229-253 (2014). <https://doi.org/10.1080/10543406.2013.860769>
- 3 Brooks, B. R. *et al.* Intravenous edaravone treatment in ALS and survival: An exploratory, retrospective, administrative claims analysis. *EClinicalMedicine* **52**, 101590 (2022). <https://doi.org/10.1016/j.eclinm.2022.101590>
- 4 van Kruijsdijk, R. C. M. *et al.* Personalizing treatment in end-stage kidney disease: deciding between haemodiafiltration and haemodialysis based on individualized treatment effect prediction. *Clin Kidney J* **15**, 1924-1931 (2022). <https://doi.org/10.1093/ckj/sfac153>
- 5 Yiu, Z. Z. N. *et al.* Randomized Trial Replication Using Observational Data for Comparative Effectiveness of Secukinumab and Ustekinumab in Psoriasis: A Study From the British Association of Dermatologists Biologics and Immunomodulators Register. *JAMA Dermatol* **157**, 66-73 (2021). <https://doi.org/10.1001/jamadermatol.2020.4202>
- 6 RUBIN, D. B. *Multiple Imputation for Nonresponse in Surveys.* (Wiley, 1987).
- 7 Chan, E. W. *et al.* Prevention of Dabigatran-Related Gastrointestinal Bleeding With Gastroprotective Agents: A Population-Based Study. *Gastroenterology* **149**, 586-595 e583 (2015). <https://doi.org/10.1053/j.gastro.2015.05.002>
- 8 Stepien, K., Nowak, K., Zalewski, J., Pac, A. & Undas, A. Extended treatment with non-vitamin K antagonist oral anticoagulants versus low-molecular-weight heparins in cancer patients following venous thromboembolism. A pilot study. *Vascul Pharmacol* **120**, 106567 (2019). <https://doi.org/10.1016/j.vph.2019.106567>
- 9 Faqah, A., Sheikh, H., Bakar, M. A., Tayyaab, F. & Khawaja, S. Comparative analysis of enoxaparin versus rivaroxaban in the treatment of cancer associated venous thromboembolism: experience from a tertiary care cancer centre. *Thromb J* **18**, 8 (2020). <https://doi.org/10.1186/s12959-020-00221-2>
- 10 Young, A. M. *et al.* Comparison of an Oral Factor Xa Inhibitor With Low Molecular Weight Heparin in Patients With Cancer With Venous Thromboembolism: Results of a Randomized Trial (SELECT-D). *Journal of clinical oncology : official journal of the American Society of Clinical Oncology* **36**, 2017-2023 (2018). <https://doi.org/10.1200/JCO.2018.78.8034>
- 11 Chen, D. Y. *et al.* Comparison Between Non-vitamin K Antagonist Oral Anticoagulants and Low-Molecular-Weight Heparin in Asian Individuals With Cancer-Associated Venous Thromboembolism. *JAMA Netw Open* **4**, e2036304 (2021). <https://doi.org/10.1001/jamanetworkopen.2020.36304>
- 12 Li, A., Manohar, P. M., Garcia, D. A., Lyman, G. H. & Steuten, L. M. Cost effectiveness analysis of direct oral anticoagulant (DOAC) versus dalteparin for the treatment of cancer associated thrombosis (CAT) in the United States. *Thromb Res* **180**, 37-42 (2019). <https://doi.org/10.1016/j.thromres.2019.05.012>
- 13 Lopes, D. G., Tamayo, A., Schipp, B. & Siepmann, T. Cost-effectiveness of edoxaban vs low-molecular-weight heparin and warfarin for cancer-associated thrombosis in Brazil. *Thromb Res* **196**, 4-10 (2020). <https://doi.org/10.1016/j.thromres.2020.08.014>
- 14 Cohen, A. T. *et al.* Patient-reported outcomes associated with changing to rivaroxaban for the treatment of cancer-associated venous thromboembolism - The COSIMO study. *Thromb Res* **206**, 1-4 (2021). <https://doi.org/10.1016/j.thromres.2021.06.021>

- 15 Picker, N. *et al.* Anticoagulation Treatment in Cancer-Associated Venous Thromboembolism: Assessment of Patient Preferences Using a Discrete Choice Experiment (COSIMO Study). *Thromb. Haemost.* **121**, 206-215 (2021). <https://doi.org/10.1055/s-0040-1714739>

REVIEWERS' COMMENTS

Reviewer #1 (Remarks to the Author):

Dear Authors,
Thanks for carefully considering all my questions and concerns regarding your submission.
I have no further comments.

With best regards, reviewer #1.

Reviewer #2 (Remarks to the Author):

No additional comments